# Invariant Image Representation Using Novel Fractional-Order Polar Harmonic Fourier Moments

**DOI:** 10.3390/s21041544

**Published:** 2021-02-23

**Authors:** Chunpeng Wang, Hongling Gao, Meihong Yang, Jian Li, Bin Ma, Qixian Hao

**Affiliations:** 1School of Computer Science and Technology (School of Cyber Security), Qilu University of Technology (Shandong Academy of Sciences), Jinan 250353, China; mpeng1122@163.com (C.W.); ghl5210808@163.com (H.G.); yangmh@sdas.org (M.Y.); ljian20@gmail.com (J.L.); haoqixian123@163.com (Q.H.); 2Shandong Provincial Key Laboratory of Computer Networks, Shandong Computer Science Center (National Supercomputer Center in Jinan), Qilu University of Technology (Shandong Academy of Sciences), Jinan 250014, China; 3Shandong Provincial Key Laboratory for Distributed Computer Software Novel Technology, Jinan 250358, China

**Keywords:** fractional-order polar harmonic Fourier moments, continuous orthogonal moments, geometric invariance, image reconstruction, object recognition

## Abstract

Continuous orthogonal moments, for which continuous functions are used as kernel functions, are invariant to rotation and scaling, and they have been greatly developed over the recent years. Among continuous orthogonal moments, polar harmonic Fourier moments (PHFMs) have superior performance and strong image description ability. In order to improve the performance of PHFMs in noise resistance and image reconstruction, PHFMs, which can only take integer numbers, are extended to fractional-order polar harmonic Fourier moments (FrPHFMs) in this paper. Firstly, the radial polynomials of integer-order PHFMs are modified to obtain fractional-order radial polynomials, and FrPHFMs are constructed based on the fractional-order radial polynomials; subsequently, the strong reconstruction ability, orthogonality, and geometric invariance of the proposed FrPHFMs are proven; and, finally, the performance of the proposed FrPHFMs is compared with that of integer-order PHFMs, fractional-order radial harmonic Fourier moments (FrRHFMs), fractional-order polar harmonic transforms (FrPHTs), and fractional-order Zernike moments (FrZMs). The experimental results show that the FrPHFMs constructed in this paper are superior to integer-order PHFMs and other fractional-order continuous orthogonal moments in terms of performance in image reconstruction and object recognition, as well as that the proposed FrPHFMs have strong image description ability and good stability.

## 1. Introduction

The rapid development of information and network technologies has brought about great changes to human life and production. As more and more digital images are conveniently transmitted and downloaded online, negative impacts also arise from such trends. Each digital image can be accessed freely, and such free access provides the opportunity for launching various attacks on digital images, such as rotation, translation, and scaling, making the application of digital watermarking [1,2] and pattern recognition [3] more difficult. To address these problems, researchers have begun to look for a kind of feature vectors that can represent the objective information contained in images, so as to figure out how to use invariant features to describe images and use a very small number of datasets to represent more image information. Image moments are a type of highly concentrated image features, which serve as a powerful tool to characterize images, and are invariant to rotation, translation, and scaling. Image moments have been widely used in various fields of image processing, including image watermarking [4], image indexing [5], face recognition [6], image registration [7], etc.

The concept of moment first appeared in the research areas of statistics and classical mechanics. In 1962, Hu first proposed Hu’s moment invariants [8] that were introduced into the field of image processing and proposed the theory of image moments used to describe image features. Later, rotational moments (RMs) [9] and complex moments (CMs) [10] were proposed successively. However, because the basis functions of rotational and complex moments are non-orthogonal, there are problems with such moments, such as their information redundancy and high sensitivity to noise, which make it difficult to reconstruct the original images using such moments. To address the challenging reconstruction problem, the concept of orthogonal moments was proposed by scholars based on the theory of orthogonal functions. Orthogonal moments are free from the problem of information redundancy; therefore, a small number of orthogonal moments can be used to easily reconstruct the original images. Due to their minimum information redundancy and high robustness [11], orthogonal moments have been used widely. Orthogonal moments include discrete and continuous orthogonal moments. In 1980, Teague [12] firstly proposed the use of Zernike moments (ZMs), a type of continuous orthogonal moments, for image description. The amplitude of ZMs is rotation invariant, and ZMs are characterized by high noise resistance and low information redundancy. Continuous orthogonal moments are invariant to rotation, scaling, and translation and are able to capture the global features of images, thus playing a great role in image reconstruction. Continuous orthogonal moments mainly include Jacobi Fourier moments (JFMs) [13], pseudo-Jacobi Fourier moments (PJFMs) [13], pseudo-Zernike moments (PZMs) [14], Gaussian-Hermite moments (GHMs) [15], Legendre moments (LMs) [12], continuous Hahn Moments (CHMs) [16], polar harmonic transforms (PHTs) [17], exponent moments (EMs) [18], Chebyshev-Fourier moments (CHFMs) [19], orthogonal Fourier-Mellin moments (OFMMs) [20], Bessel-Fourier moments (BFMs) [21], radial harmonic Fourier moments (RHFMs) [22], polar harmonic Fourier moments (PHFMs) [23], etc. Owing to their high numerical stability, PHFMs are superior to other continuous orthogonal moments in terms of performance in image reconstruction and object recognition.

However, existing orthogonal moments are limited to the integer-order and now there are very few studies on non-integer-order orthogonal moments. In recent years, fractional-order problems, such as fractional-order calculus and fractional-order Fourier transform [18], have attracted extensive attention and more and more researchers have begun to put focus on fractional-order moments. Xiao et al. derived the fractional-order Legendre-Fourier moments (FrOLFMs) [24]. Zhang et al. defined the fractional-order Fourier-Mellin polynomial and then derived the fractional-order orthogonal Fourier-Mellin moments (FrOFMMs) [25]. Benouini et al. and Yang et al. proposed the orthogonal fractional-order Chebyshev moments (FrOCMs) [26] and the fractional-order Zernike moments (FrZMs) [27], respectively. Chen et al. introduced the quaternion orthogonal fractional-order Zernike moments (QFrZMs) [28] used to process color images. Hosny et al. proposed the fractional-order polar harmonic transforms (FrPHTs) [29] and the multi-channel fractional-order radial harmonic Fourier moments (FrMRHFMs) [30]. Among the aforementioned integer-order continuous orthogonal moments, PHFMs have strong image description ability and can deliver superior performance in image reconstruction and object recognition. Therefore, in this paper, the idea of fractional order is incorporated into PHFMs, fractional-order radial polynomials are constructed by modifying the integer-order radial polynomials of PHFMs to extend the traditional PHFMs to fractional polar harmonic Fourier moments (FrPHFMs), then the properties of FrPHFMs are analyzed in detail, and finally, it is experimentally verified that the proposed FrPHFMs have better performance than integer-order PHFMs and other fractional-order continuous orthogonal moments in image reconstruction and object recognition.

The main contributions of the study are summarized below. (1) Integer-order PHFMs, which can only take integer numbers, are extended to FrPHFMs by means of modification of their radial polynomials. (2) The relationship between the changes in radial polynomials and the reconstructed images is identified by analyzing the rate of change of radial polynomials, and it is experimentally verified that the FrPHFMs constructed using the proposed algorithm have good performance in image reconstruction and noise resistance. (3) The constructed FrPHFMs are used for object recognition and compared with integer-order PHFMs and other fractional-order continuous orthogonal moments from the perspective of performance. The results of comparison show that the proposed FrPHFMs have better performance in image reconstruction and object recognition.

Other sections of this paper are organized as follows: Section 2 introduces the FrPHFMs construction process in detail and analyzes the geometric invariance of FrPHFMs; Section 3 mainly analyzes the properties of FrPHFMs from two perspectives, namely the changes in, and the rate of change of, their radial polynomials; Section 4 describes in detail the experiments and discussions with respect to image reconstruction, geometric invariance, and object recognition; and Section 5 draws a conclusion of this study.

## 2. FrPHFMs

In this section, the construction and properties of FrPHFMs are described in detail. Firstly, the traditional integer-order PHFMs are introduced, then the definition of FrPHFMs is given, and, finally, the geometric invariance of FrPHFMs is discussed.

### 2.1. Definition of Integer-Order PHFMs

The integer-order PHFMs with order of n≥0 and repetition of $\left|m\right|\geq 0$ of image f(r,θ) in a polar coordinate system [31] is defined as:(1)$$p_{nm}=\frac{2}{\pi }\int_{0}^{2\pi }\int_{0}^{1}f(r,\theta )\bar{H_{nm}(r,\theta )}rdrd\theta$$
where $\bar{\left[\cdot \right]}$ is the conjugate of a complex number, and basis function $H_{nm}(r,\theta )$ is composed of radial polynomial $T_{n}(r)$ and angular Fourier factor exp(jmθ):(2)$$H_{nm}(r,\theta )=T_{n}(r)\exp(jm\theta )$$
where radial polynomial $T_{n}(r)$ is
(3)$$T_{n}(r)=\left\{\begin{array}{ll} 1/\sqrt{2} & \mathrm{while}\;n\;\mathrm{is}\;0\\ \sin((n+1)\pi r^{2}) & \mathrm{while}\;n\;\mathrm{is}\;\mathrm{odd}\\ \cos(n\pi r^{2})\; & \mathrm{while}\;n\;\mathrm{is}\;\mathrm{even} \end{array}\right.$$

$T_{n}(r)$ is orthogonal within the range of 0≤r≤1:(4)$$\int_{0}^{1}T_{n}(r)T_{n^{\prime }}(r)rdr=\frac{1}{4}\delta_{nn^{\prime }}$$

From the property of angular Fourier factor exp(jmθ) and the formula above, it can be known that basis function $H_{nm}(r,\theta )$ is orthogonal in the unit circle [32]:(5)$$\int_{0}^{2\pi }\int_{0}^{1}H_{nm}(r,\theta )\bar{H_{kl}(r,\theta )}rdrd\theta =\frac{\pi }{2}\delta_{nk}\delta_{ml}$$
where 0≤r≤1, 0≤θ≤2π, and δ is the Kronecker delta.

According to the theory of complete system of orthogonal functions, original image function f(r,θ) can be approximately reconstructed using a finite number of PHFMs. Given the PHFMs with the maximum order of $n_{\max}$ and the maximum repetition of $m_{\max}$, the formula for approximate reconstruction of the original image is
(6)$$f(r,\theta )\approx \sum_{n=0}^{n_{\max}}\sum_{m=-m_{\max}}^{m_{\max}}P_{nm}H_{nm}(r,\theta )$$

### 2.2. Definition of FrPHFMs

In this paper, integer-order PHFMs are extended to FrPHFMs. The key to the construction of FrPHFMs is to construct the fractional-order radial polynomials; therefore, the extension of orthogonal radial polynomial $T_{n}(r)$ is considered. Letting t>0, $U_{n}^{\left(t\right)}(r)$ be defined as the radial polynomial below:(7)$$U_{n}^{\left(t\right)}(r)=\left\{\begin{array}{ll} 1/\sqrt{2} & \mathrm{while}\;n\;\mathrm{is}\;0\\ \sqrt{t}r^{t-1}\sin((n+1)\pi r^{2t})\; & \mathrm{while}\;n\;\mathrm{is}\;\mathrm{odd}\\ \sqrt{t}r^{t-1}\cos(n\pi r^{2t})\; & \mathrm{while}\;n\;\mathrm{is}\;\mathrm{even} \end{array}\right.$$
then the basis function of the FrPHFMs is:(8)$$W_{nm}^{\left(t\right)}(r,\theta )=U_{n}^{\left(t\right)}(r)\exp(jm\theta )$$

$U_{n}^{\left(t\right)}(r)$ complies with the following orthogonal relationship within the range of 0≤r≤1:

(9)$$\int_{0}^{1}U_{n}^{\left(t\right)}(r)U_{n}^{\prime (t)}(r)rdr=\int_{0}^{1}r^{t}U_{n}^{\left(t\right)}(r^{t})U_{n}^{\prime (t)}(r^{t})d(r^{t})=\frac{1}{4}\delta_{nn^{\prime }}$$

From the properties of the angular Fourier factor and radial polynomials, it can be known that basis function $W_{nm}^{\left(t\right)}(r,\theta )$ is orthogonal in the unit circle and complies with the following orthogonal relationship:

(10)$$\begin{array}{l} \int_{0}^{2\pi }\int_{0}^{1}W_{nm}^{\left(t\right)}(r,\theta )\bar{W_{kl}^{\left(t\right)}(r,\theta )}rdrd\theta \\ =\int_{0}^{2\pi }\int_{0}^{1}U_{nm}^{\left(t\right)}(r)\exp(jm\theta ){U^{\prime }}_{kl}^{\left(t\right)}(r)\exp(-jl\theta )rdrd\theta \\ =\frac{\pi }{2}\delta_{nk}\delta_{ml} \end{array}$$

Given t>0, the FrPHFMs with an order of n≥0 and a repetition of $\left|m\right|\geq 0$ is defined as:(11)$$FP_{nm}^{\left(t\right)}=\frac{2}{\pi }\int_{0}^{2\pi }\int_{0}^{1}f(r,\theta )\bar{W_{nm}^{\left(t\right)}(r,\theta )}rdrd\theta$$

From the formula above, it can be seen that, when t=1, the FrPHFMs is an integer-order PHFMs, and thus is an extension to the integer-order PHFMs.

Given the FrPHFMs with the max moment order of $n_{\max}$ and the maximum repetition of $m_{\max}$, the original image can be approximately reconstructed using the formula below:(12)$$f(r,\theta )\approx \sum_{n=0}^{n_{\max}}\sum_{m=-m_{\max}}^{m_{\max}}FP_{nm}^{\left(t\right)}W_{nm}^{\left(t\right)}(r,\theta )$$

### 2.3. Geometric Invariance of FrPHFMs

**Property** **1.**Rotation invariance of FrPHFMs

The amplitudes of FrPHFMs are invariant to image rotation. Assuming that f(r,θ) is an image function in a polar coordinate system, its FrPHFMs is $FP_{nm}$, image f(r,θ+φ) is obtained by rotating the original image φ degree and the FrPHFMs thereof is $F{P^{\prime }}_{nm}$, then according to the calculation formula, the FrPHFMs in the polar coordinate system is:(13)$$\begin{array}{cc} F{P^{\prime }}_{nm} & =\frac{2}{\pi }\int_{0}^{2\pi }\int_{0}^{1}f(r,\theta +\varphi )U_{n}^{\left(t\right)}(r)\exp(-jm\theta )rdrd\theta \\ & =\frac{2}{\pi }\int_{0}^{2\pi }\int_{0}^{1}f(r,\theta )U_{n}^{\left(t\right)}(r)\exp(-jm(\theta -\varphi ))rdrd\theta \\ & =\frac{2}{\pi }\int_{0}^{2\pi }\int_{0}^{1}f(r,\theta )U_{n}^{\left(t\right)}(r)\exp(-jm\theta )rdrd\theta \exp(jm\varphi )\\ & =FP_{nm}\exp(jm\varphi ) \end{array}$$

The amplitudes are taken on both sides of the equation above:(14)$$\left|F{P^{\prime }}_{nm}\right|=\left|FP_{nm}\exp(jm\varphi )\right|=\left|FP_{nm}\right|\left|\exp(jm\varphi )\right|=\left|FP_{nm}\right|$$

From the formula above, it can be known that the amplitudes of the FrPHFMs of the image obtained by rotating the original image are equal to those of the FrPHFMs of the original image, indicating that FrPHFMs are invariant to image rotation. In this way, the angle of rotation, φ, can also be estimated by comparing the moments of the two images.

**Property** **2.**Scaling invariance of FrPHFMs

When calculating the scaled FrPHFMs, for a given image function $g(r^{\prime },\theta )$, find the k of the image radius, the range of variation of $r^{\prime }$ will be $0\leq r^{\prime }\leq k$, and the normalized image function will be:(15)$$g(r^{\prime },\theta )=g(kr,\theta )=f(r,\theta )$$ where the variation range of $r=\frac{r^{\prime }}{k}$ is 0≤r≤1. f(r,θ) is the normalized image function, and the FrPHFMs calculated by using the normalized function f(r,θ) has scale invariance. Because any image $f(\frac{r^{\prime }}{k},\theta )$ obtained by scaling the same image function f(r,θ) is finally normalized to the same function f(r,θ) according to Formula (15), the normalized image FrPHFMs has scaling invariance.

## 3. Analysis of Radial Polynomials

Although continuous orthogonal moments have good image description ability, they will be affected by various errors and numerical instability under the condition of high order, and these factors will affect their accuracy. Because such errors have negative effect on image analysis and reconstruction, the image reconstruction performance of continuous orthogonal moments will become very poor when their order reaches the critical value. The properties of continuous orthogonal moments are mainly reflected in their radial polynomials. In this section, the properties of radial polynomials are analyzed. Two groups of test images are shown in the figures below.

Figure 1 shows the plots of FrPHFMs radial polynomials versus r with the order n=85 and different t values, where the value range of r is 0≤r≤1. When t=1, FrPHFMs will be PHFMs. It can be seen from Figure 1 that the radial polynomial variation of integer-order PHFMs is unstable, while the radial polynomial variation of FrPHFMs is relatively stable. It can also be observed in Figure 1 that the radial polynomials of integer-order PHFMs change rapidly near the point where r=1, resulting in numerical instability near this point. When t=0.7,0.8,0.9, the radial polynomials of FrPHFMs change in a relatively stable manner, effectively reducing the errors occurring in the edge of images reconstructed with PHFMs.

To show the changing degree of the radial polynomial versus r more clearly, we calculate the derivative of the radial polynomial to r as the change rate of FrPHFMs radial polynomial. Figure 2 shows the change rate of the corresponding radial polynomials with the increase of r. It can be seen from Figure 2 that the change rate of radial polynomials of integer-order PHFMs is generally higher than the FrPHFMs with the same order near r=1. And the radial polynomials of PHFMs change rapidly, resulting in numerical instability in the edge regions of images [33] and very poor image reconstruction results. However, when t=0.7,0.8,0.9, the change of radial polynomials near r=1 is relatively stable, and the change rate is small, thereby, realizing greatly improved image reconstruction results. When the radial polynomials change too fast, the radial polynomials oscillates around r−axis at a higher frequency, which leads to the fact that the radial polynomials cannot be correctly represented by a single value at the pixel center, resulting in poor reconstruction effect and unclear reconstructed image. On the contrary, when the value of t is fractional parameter, the changes in the radial polynomials of FrPHFMs are stable, indicating that FrPHFMs can achieve clear display effects of reconstructed images, reduce experimental errors, and effectively mitigate the deficiency of PHFMs.

## 4. Experiments and Analysis of Experimental Results

In this section, the performance of FrPHFMs in image reconstruction and object recognition, as well as their geometric invariance, are tested through a series of experiments. Thirty grayscale images with a size of 128 × 128 were used as the test images in these experiments. Figure 3 shows 10 grayscale images randomly selected from the USC-SIPI Image Database.

### 4.1. Image Reconstruction

Image reconstruction performance is an important feature of the continuous orthogonal moments [34] and reflects the accuracy of such moments in image reconstruction. When FrPHFMs are used to reconstruct images, the computational efficiency can be improved by limiting the number of FrPHFMs involved in image reconstruction. In other words, given $n+\left|m\right|\leq K\;(0\leq n\leq n_{\max},0\leq \left|m\right|\leq m_{\max})$, where K is a constant [31], the formula for image reconstruction using FrPHFMs can be written as:(16)$$f(r,\theta )=\sum_{n=0}^{n_{\max}}\sum_{m=-m_{\max}}^{m_{\max}}FP_{nm}^{\left(t\right)}W_{nm}^{\left(t\right)}(r,\theta ),n+\left|m\right|\leq K$$

Similarly, other types of moments also are subject to reconstruction constraints. The reconstruction constraints and the numbers of moments used for image reconstruction for three types of moments are listed in Table 1.

#### 4.1.1. Comparison between FrPHFMs and Integer-Order PHFMs

**Experiment 1.** A grayscale image named Lena with a size of 128 × 128 pixels was used for this experiment, the max moment order $n_{\max}$ was set to varying number from 50 to 90 at an interval of 5, and t=0.7,0.8,0.9,1.0,1.1,1.2. The number of moments used to reconstruct the original image is 2601, 3136, 3721, 4356, 5041, 5776, 6561, 7396 and 8281 respectively. Unless otherwise specified, the value of K in this paper is $K=n_{\max}$.The experimental results are summarized in Table 2. For the purpose of comparison, the results of image reconstruction with integer-order PHFMs (t=1) are also listed in Table 2, in which ε denotes the mean square reconstruction error (MSRE) [14], which is defined as: (17)$$\varepsilon =\frac{\sum_{x=0}^{M-1}\sum_{y=0}^{N-1}{\left|f(x,y)-\bar{f}(x,y)\right|}^{2}}{\sum_{x=0}^{M-1}\sum_{y=0}^{N-1}f^{2}(x,y)}$$ where f(x,y) denotes the original image with a size of M×N, and $\bar{f}(x,y)$ denotes the reconstructed image with the same size.

It can be seen from the data in Table 2 that, within a certain order range, with the increase of $n_{\max}$, the reconstruction effect of FrPHFMs becomes better, the reconstructed image becomes clearer, and the error becomes smaller. When it exceeds this range, the error will gradually increase, and the reconstructed image will become blurred, which is caused by the instability of the radial polynomial. When $n_{\max}$ is about 70, with the increase of the order, the reconstruction error of FrPHFMs with t=0.8,0.9 shows a trend of overall decrease compared with that of integer-order PHFMs. At the same time, with the increase of $n_{\max}$, the edge part of the reconstructed image with the same t value will appear unclear region, which can be explained by the change of radial polynomial. The change of fractional orthogonal moments is mainly realized by changing the radial polynomial. For the same order, the radial polynomial can be adjusted by changing the value of fractional parameter t, which slows down the appearance of white regions and then changes the image reconstruction effect.

Subsequently, the reconstruction errors produced by FrPHFMs were compared with those by PHFMs (t=1), and the mean value of reconstruction errors was taken for the 30 images. The experimental results are shown in Figure 4, in which reconstruction errors are measured by MSRE. The line chart above clearly shows that, when $n_{\max}$ is greater than 70, the reconstruction errors with respect to the images processed with integer-order PHFMs become greater rapidly, and, when t<1, the reconstruction errors produced by FrPHFMs are remarkably smaller than those by integer-order PHFMs, indicating that FrPHFMs can effectively mitigate the trend of overall increase in image reconstruction errors.

**Experiment 2.** Considering the serious effect of noise on image reconstruction [35], the trend of change in reconstruction errors with the increase in order was tested when different values of t were taken after the same salt and pepper noises were added. Thirty images were used for this experiment, the mean value of reconstruction errors was taken for the 30 images, $n_{\max}$ was set to varying number within the range from 10 to 100, t=0.7,0.8,0.9,1.0, and salt and pepper noises at intensity levels of 0.05, 0.1, 0.15, and 0.2 were added, respectively. The line chart for image reconstruction errors is shown in Figure 5.

The results of this experiment show that, as the density of added salt and pepper noise increases, the image reconstruction effects will deteriorate, indicating that noise does affect image reconstruction. When t<1, the reconstruction effects of FrPHFMs are always better than those of integer-order PHFMs. When $n_{\max}$ is greater than 85, the reconstruction errors produced by PHFMs become greater rapidly, while the reconstruction errors produced by FrPHFMs are smaller than those by integer-order PHFMs, indicating that FrPHFMs can effectively mitigate the trend of overall increase in image reconstruction errors.

#### 4.1.2. Comparison between FrPHFMs and Other Fractional-Order Continuous Orthogonal Moments

In this subsection, FrPHFMs are compared with FrRHFMs and FrEMs from the perspective of image reconstruction effects. Different values of t are required to allow different fractional-order moments to achieve good image reconstruction effects. The results of many experiments indicate that the optimal reconstruction effects achievable by three different types of moments correspond to different values of t. Therefore, the image reconstruction performance of FrPHFMs (t=0.7,0.8,0.9,1.0) was compared with that of FrRHFMs (t=1.2,1.3,1.4,1.0) and FrEMs (t=1.3,1.4,1.5,1.0), and the max moment order was set to $n_{\max}=50,55,\ldots ,75$, respectively. The results of comparison of images reconstructed with different moments are summarized in Table 3.

In order to demonstrate image reconstruction errors more visually, a line chart was drawn to allow for comparison of the reconstruction errors produced by the three different moments. Curves in different colors represent the reconstruction errors produced by different moments for certain values of t, as shown in Figure 6. From the line chart above, it can be seen that the reconstruction errors produced by FrPHFMs are very small and are much smaller than those produced by FrEMs; and, as $n_{\max}$ increases continuously, the reconstruction errors produced by the three different moments become greater gradually, but those produced by FrPHFMs become greater more slowly, indicating that FrPHFMs can effectively mitigate the trend of increase in image reconstruction errors. These results show that FrPHFMs are superior to FrRHFMs and FrEMs in terms of image reconstruction performance.

### 4.2. Experiments on Geometric Invariance

The amplitudes of FrPHFMs are invariant to rotation and scaling. In other words, the amplitudes of the FrPHFMs of the rotated and scaled images will be approximately equal to those of the FrPHFMs of the original images. The rotation invariance and scaling invariance of FrPHFMs are experimentally demonstrated in this section.

(1)Rotation invariance

The amplitudes of the FrPHFMs of a group of rotated images were calculated. A grayscale image named Lena with a size of 128 × 128 pixels was used for this experiment. After the image was rotated 5°, 15°, 25°, 35°, and 45°, respectively, when t=0.7,0.8,0.9, the amplitudes of the FrPHFMs of the rotated images were calculated, respectively, and compared with those of the FrPHFMs of the original image. Figure 7 shows the rotated versions of image Lena, and the experimental results are summarized in Table 4, Table 5 and Table 6 below.

The amplitudes of 9 moments of the rotated images and the original image when different values of t are taken are given in the tables above. Through comparison of the listed data, it is found that the amplitudes of the same FrPHFMs of all rotated images are approximately equal, indicating that the amplitudes of FrPHFMs are invariant to rotation [36].

(2)Scaling invariance

The FrPHFMs of a group of scaled images were calculated to verify the scaling invariance thereof. A grayscale image named Lena with a size of 128 × 128 pixels was used for this experiment. After the image was scaled by 0.5, 0.7, 1.25, and 1.5 times, respectively, the amplitudes of the FrPHFMs of the scaled images were calculated, respectively, and compared with those of the FrPHFMs of the original image. Figure 8 shows the scaled versions of image Lena, and the experimental results are summarized in Table 7, Table 8 and Table 9 below.

The amplitudes of 9 moments of the scaled images and the original image when different values of t are taken are given in the tables above. Through comparison of the listed data, it is found that the amplitudes of the same FrPHFMs of all scaled images are approximately equal, indicating that the amplitudes of FrPHFMs are invariant to scaling.

### 4.3. Object Recognition 

In this section, the proposed FrPHFMs are experimentally compared with other fractional-order continuous orthogonal moments, including fractional-order polar complex exponential transforms (FrPCETs), fractional polar cosine transforms (FrPCTs), fractional polar sine transforms (FrPSTs), and FrZMs, from the perspective of object recognition. Firstly, relevant operations were performed on the test images, the moments of tested images were calculated, and these moments were used as image features. Because errors will occur when orthogonal moments are calculated, accurate moments need to be selected and used for comparative experiments on object recognition in order to ensure accurate object recognition. The sets of different fractional-order orthogonal moments and the number of moments in each set used for object recognition are listed in Table 10. It is to be noted that low-order moments are used as image features for comparative experiments on object recognition because such moments can represent more image information using less data. 

**Experiment 1.** In this experiment, the binary images of 26 capitalized English letters were used for object recognition, and the relationship between the number of moments used for object recognition and correct classification percentage (CCP) was tested. The size of each image was 64 × 64 pixels, as shown in Figure 9. Firstly, each of the 26 letters was rotated 0−180° at an interval of 5°, producing 37 rotated images, from which 19 images were selected randomly as training images, and the remaining 18 images were used as test images. Nine hundred and sixty-two test images were produced in total, including 494 images in the training set and 468 images in the test set. Gaussian noise with average intensity of 0 and variance of $\sigma^{2}=0.20$ was added to the images in the test set, which were classified using the K-Nearest Neighbors (KNN) algorithm [37]. The changes in the CCP of different fractional-order continuous orthogonal moments with changing number of moments are shown in Figure 10 below.

From the figure above, it can be seen that the CCP of FrPHFMs is higher than those of other fractional-order continuous orthogonal moments, indicating that FrPHFMs are superior to other fractional-order continuous orthogonal moments in terms of classification performance. Additionally, it can also be observed that, as the number of moments increases, the CCP will increase gradually, and it will start to decrease after reaching a certain value, which is known as the overfitting problem in machine learning.

**Experiment 2.** One hundred grayscale images with an equal size of 128 × 128 pixels selected from database COIL-100 were used as the test images for this experiment. Figure 11 shows some images randomly selected from this database. Firstly, each image was rotated 0−90° at an interval of 5°, producing 19 images, from which 10 images were randomly selected as training images, and the remaining 9 images were used as test images. One thousand nine hundred test images were produced in total, including 1000 images in the training set and 900 images in the test set. Salt and pepper noises with varying density of $\sigma^{2}=0.00,0.05,\cdots ,0.25$ were added to the images in the test set, which were classified using the multi-class naive Bayes classifier [38]. Six, 18, 45 and 96 low-order moments were selected and used, respectively, as the image features for this experiment, and the CCPs of these different moments were compared. The experimental results are shown in Figure 12.

From Figure 12, it can be clearly observed that the CCP of FrPHFMs is higher than those of other fractional-order continuous orthogonal moments, indicating that FrPHFMs have the best classification performance. Additionally, it can also be observed that, when nose is not added to the images, the CCP of each moment is 100% but will decline with the increase in noise density. However, the CCP of FrPHFMs decreases very slowly, indicating that FrPHFMs can effectively mitigate the trend of rapid decline of object recognition rate.

**Experiment 3.** The 100 grayscale images used in experiment 2 were used as the training images for this experiment. Various conventional attacks were applied to the images, and the object recognition rates of different moments of images subjected to such attacks were compared. Firstly, all images were rotated 0°,5°,…,90°, respectively, producing 1900 images in total, which were included into the test set. Then different attacks were applied to the test images. The applied attacks included JPEG compression with quality factor of 10, 20, 50, and 80, Wiener filtering with a window size of 2 × 2 and Gaussian filtering with a window size of 4 × 4, contrast enhancement filtering with alpha factor of 0.2 and 0.4, and circular averaging filtering at radius of 2 and 4. The images in the test set were then classified using a multi-class support vector machine (MSVM) [39]. Five and 13 low-order moments were selected and used, respectively, as the image features for this experiment, and the CCPs of these different moments were compared. The experimental results are shown in Table 11.

From the Table 11, it can be seen that the CCPs of all fractional-order continuous orthogonal moments are 100% under various attacks because such moments are highly robust to attacks. It can also be observed that, regardless of which attack is applied to the images, the CCP of FrPHFMs is always higher than those of other fractional-order continuous orthogonal moments, indicating that FrPHFMs are highly robust and superior to other fractional-order continuous orthogonal moments in terms of resistance to attacks.

## 5. Conclusions

In this paper, in order to improve the anti-noise and reconstruction performance of PHFMs, the traditional PHFMs, which can only take integer-order, are extended to FrPHFMs. By modifying the radial polynomial of integer-order PHFMs, FrPHFMs are constructed according to fractional radial polynomial, and the properties of FrPHFMs are introduced and detailed experiments are carried out according to their properties. Firstly, the traditional integer-order PHFMs are introduced, and then FrPHFMs are constructed by using fractional radial polynomial, and their properties are described in detail. FrPHFMs have good orthogonality, rotation invariance, and scaling invariance and are superior to integer-order PHFMs. Secondly, the change of radial polynomial is analyzed in detail. Finally, the constructed FrPHFMs are applied to image reconstruction, geometric invariance, and object recognition experiments, which further verifies their good geometric invariance and image description ability. From the numerical and experimental analysis, the following conclusions can be drawn: FrPHFMs not only maintain the orthogonality, rotation invariance, and scaling invariance of integer-order PHFMs, but they also have good image description ability. Their performance in image reconstruction, anti-noise performance, and object recognition is better than integer-order PHFMs and other fractional-order continuous orthogonal moments. In the future, the improvement of FrPHFMs performance will be made an important area of research.

## Figures and Tables

**Figure 1 sensors-21-01544-f001:**
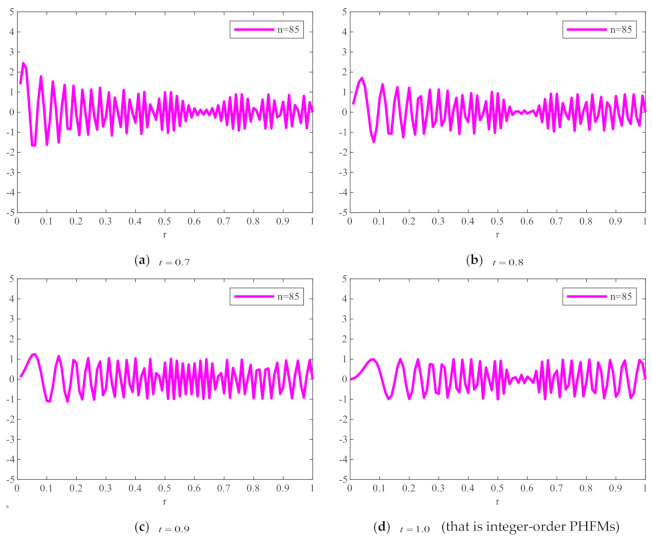
Radial polynomials of fractional-order polar harmonic Fourier moments (FrPHFMs) with the same moment order (*n* = 85) and different *t* values.

**Figure 2 sensors-21-01544-f002:**
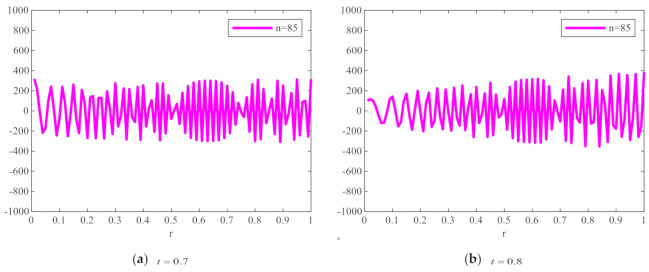
The change rate of FrPHFMs radial polynomial with the same moment order (*n* = 85) and dif-ferent *t* values.

**Figure 3 sensors-21-01544-f003:**
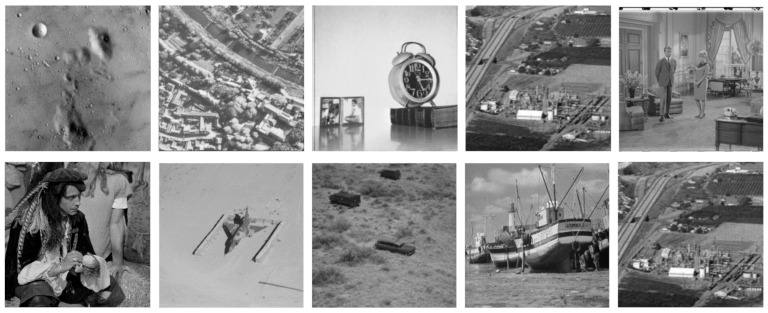
Ten of grayscale images used in the experiments.

**Figure 4 sensors-21-01544-f004:**
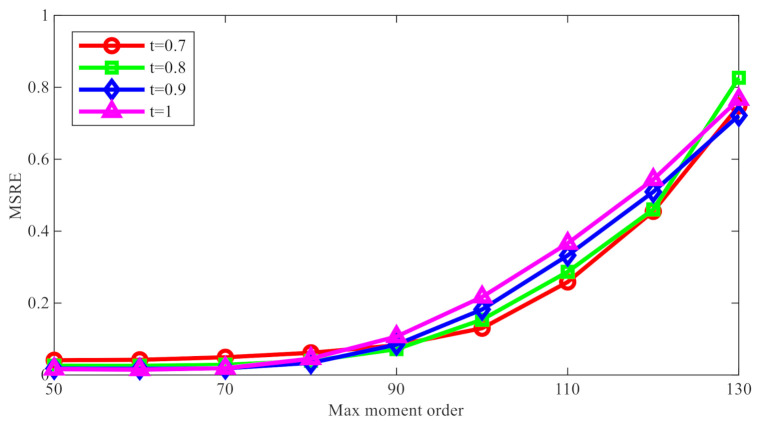
Comparison of image reconstruction errors under different t values.

**Figure 5 sensors-21-01544-f005:**
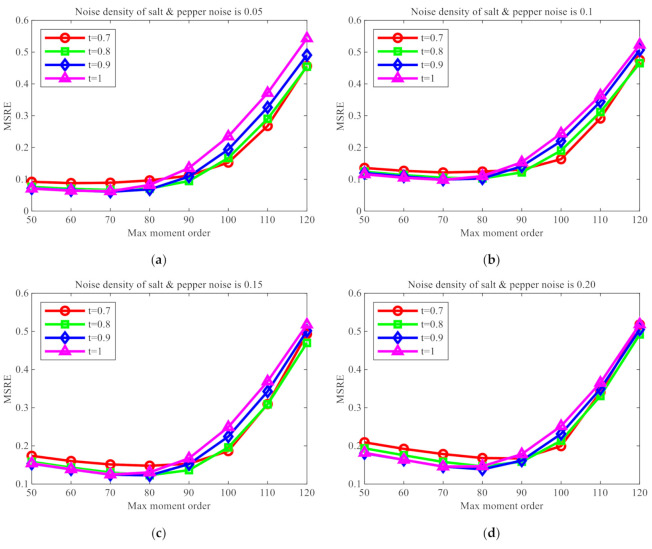
Image reconstruction error comparison after adding salt and pepper noise with different intensities.

**Figure 6 sensors-21-01544-f006:**
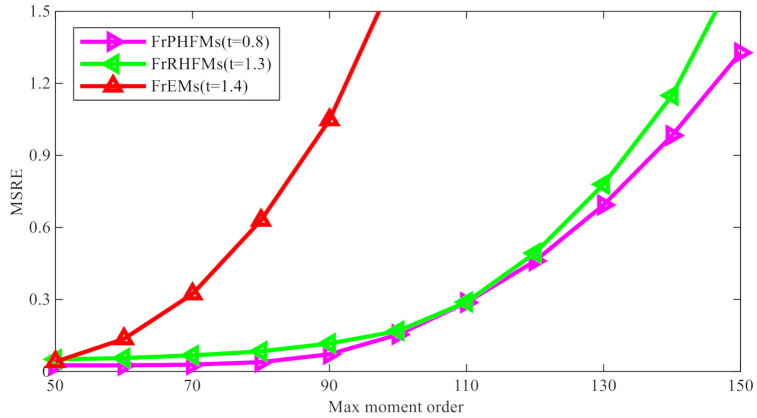
Comparison of image reconstruction errors between FrPHFMs, FrRHFMs, and FrEMs.

**Figure 7 sensors-21-01544-f007:**
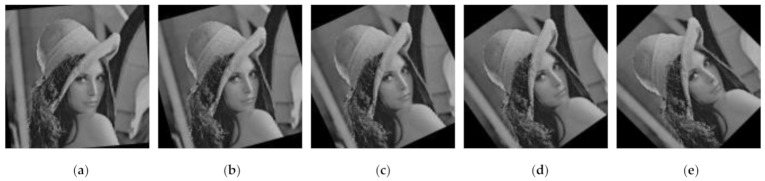
Rotated Lena (**a**) Rotation 5°, (**b**) Rotation 15°, (**c**) Rotation 25°, (**d**) Rotation 35°, and (**e**) Rotation 45°.

**Figure 8 sensors-21-01544-f008:**
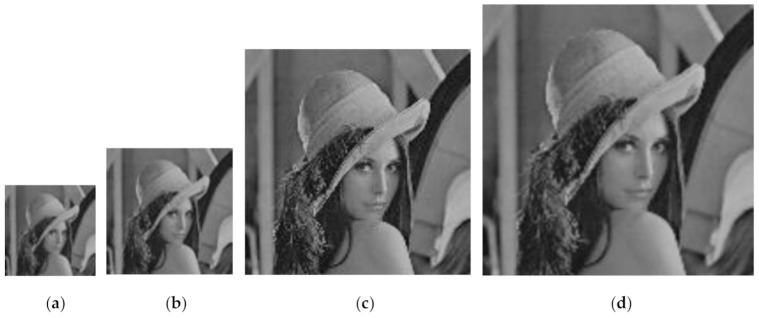
Scaled Lena. (**a**) Scaling 0.5. (**b**) Scaling 0.7. (**c**) Scaling 1.25. (**d**) Scaling 1.5.

**Figure 9 sensors-21-01544-f009:**
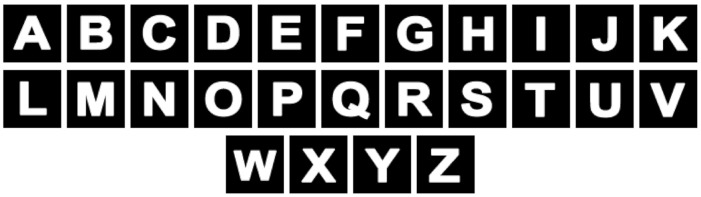
Capital images.

**Figure 10 sensors-21-01544-f010:**
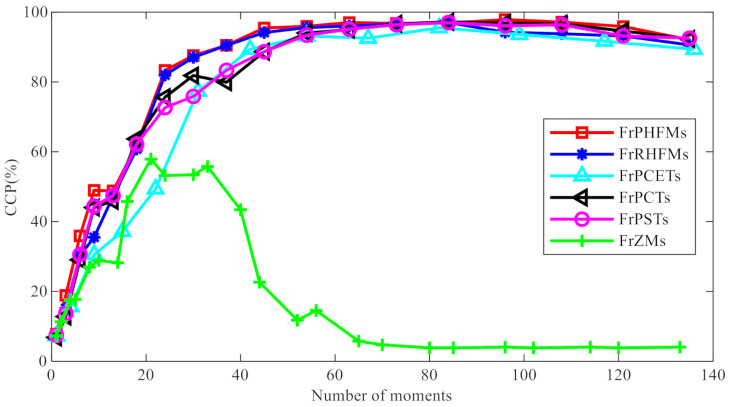
Correct classification percentage (CCP) (%) comparison of different fractional-order continuous orthogonal moments with the number of moments.

**Figure 11 sensors-21-01544-f011:**
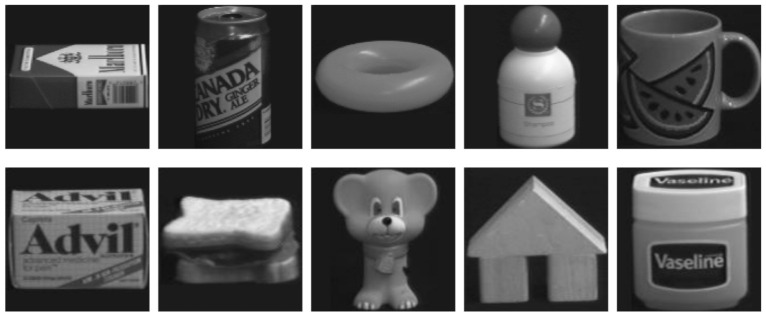
Some images in COIL-100 database.

**Figure 12 sensors-21-01544-f012:**
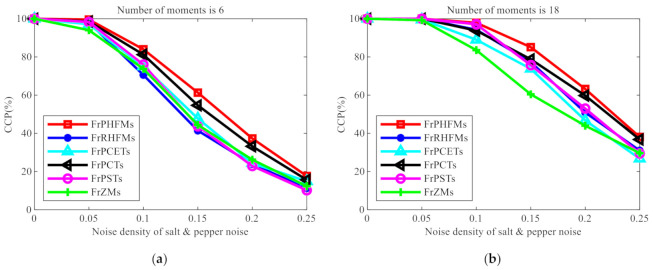
CCP (%) comparison of different fractional-order continuous orthogonal moments with the density of salt and pepper noise.

**Table 1 sensors-21-01544-t001:** Reconstruction constraints and the numbers of moments used for image reconstruction.

Moment	Limit Condition	Number of Moments
FrPHFMs	$n+\left|m\right|\leq K$	${\left(K+1\right)}^{2}$
FrRHFMs	$n+\left|m\right|\leq K$	${\left(K+1\right)}^{2}$
FrEMs	$\left|n\right|+\left|m\right|\leq K$	${\left(K+1\right)}^{2}+K^{2}$

**Table 2 sensors-21-01544-t002:** Comparison of reconstructed images of FrPHFMs with different t.

$n_{\max}$	t					
	**0.7**	**0.8**	**0.9**	**1**	**1.1**	**1.2**
50	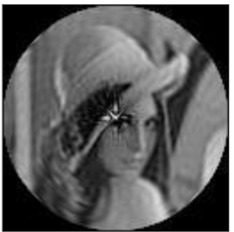	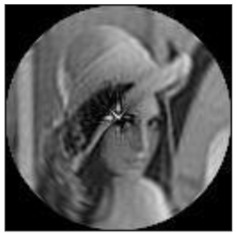	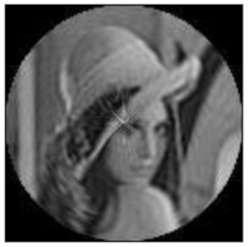	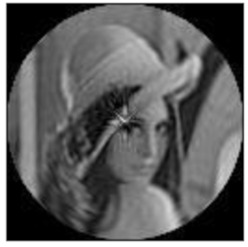	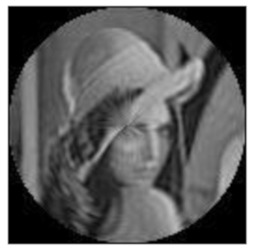	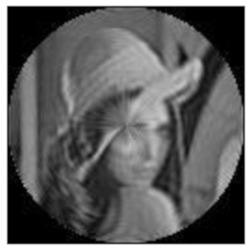
ε	0.0410	0.0258	0.0180	0.0162	0.0185	0.0243
60	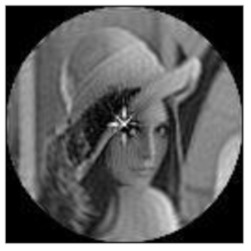	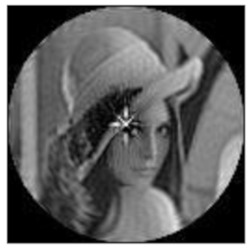	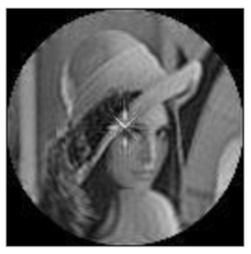	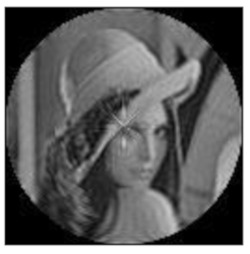	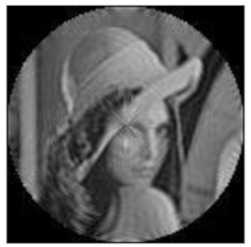	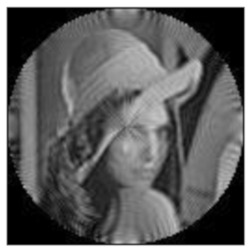
ε	0.0422	0.0255	0.0172	0.0147	0.0177	0.0256
70	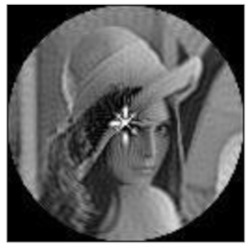	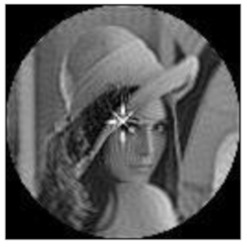	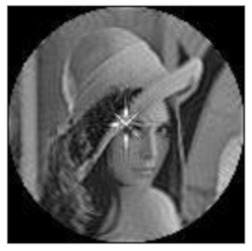	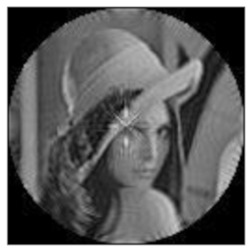	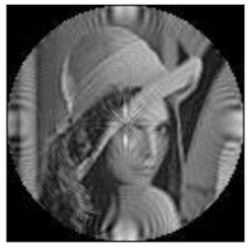	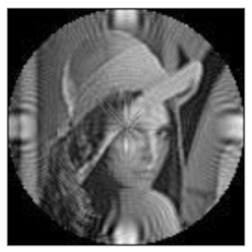
ε	0.0493	0.0285	0.0186	0.0189	0.0286	0.0435
80	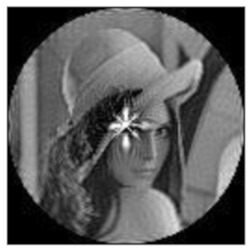	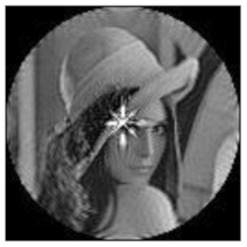	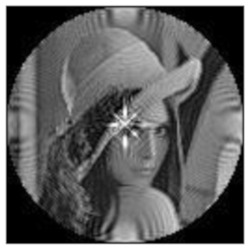	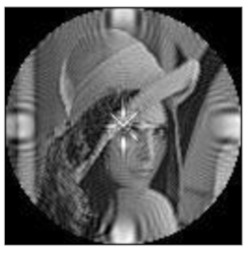	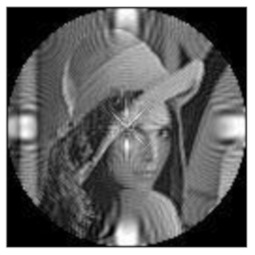	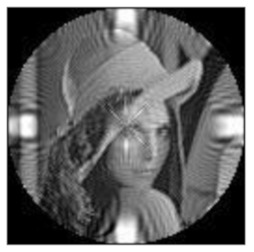
ε	0.0617	0.0390	0.0336	0.0455	0.0656	0.0883
90	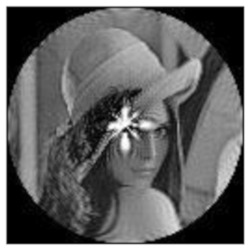	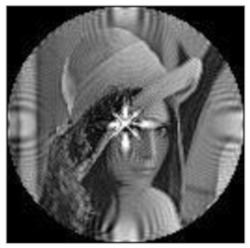	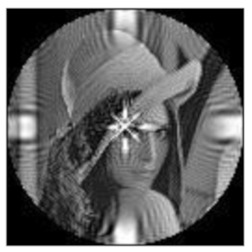	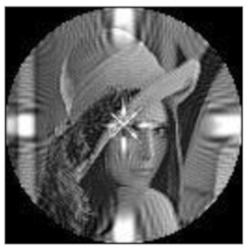	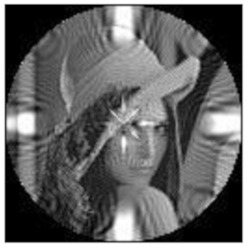	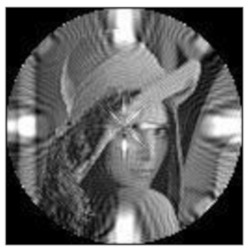
ε	0.0827	0.0717	0.0845	0.1069	0.1357	0.1695

**Table 3 sensors-21-01544-t003:** Comparison of image reconstruction between FrPHFMs, fractional-order radial harmonic Fourier moments (FrRHFMs), and FrEMs.

$n_{\max}$		**50**	**55**	**60**	**65**	**70**	**75**
FrPHFMs	0.7	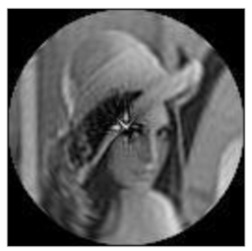 0.0410	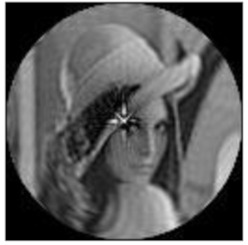 0.0408	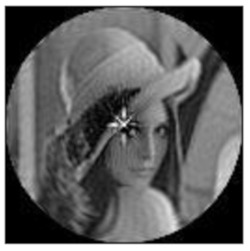 0.0422	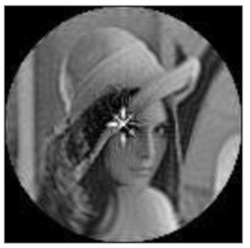 0.0447	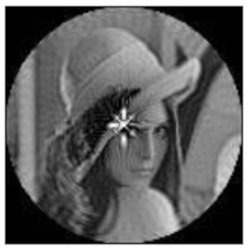 0.0493	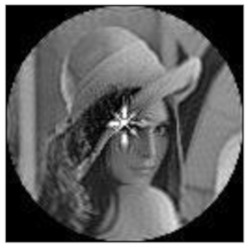 0.0552
	0.8	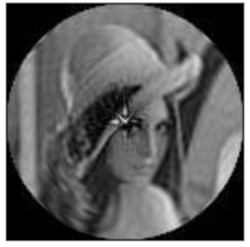 0.0258	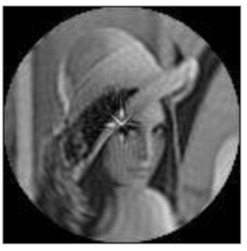 0.0254	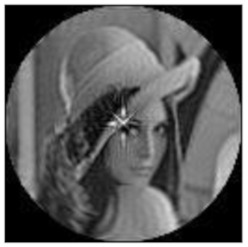 0.0255	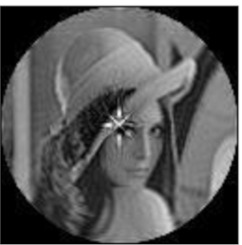 0.0261	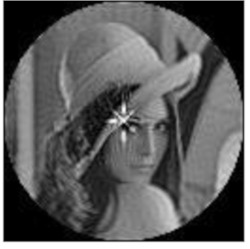 0.0285	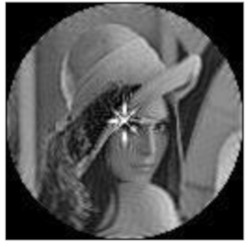 0.0322
	0.9	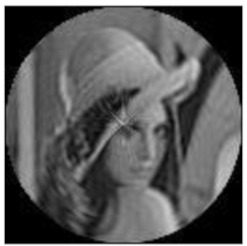 0.0180	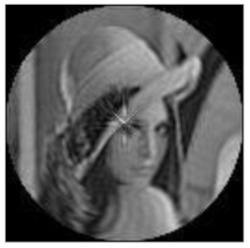 0.0173	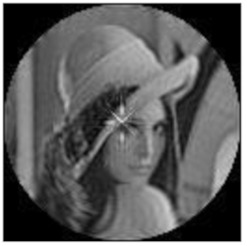 0.0172	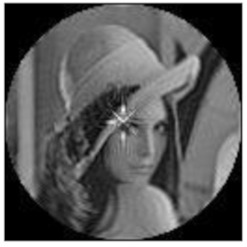 0.0174	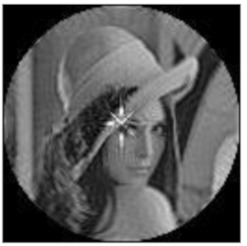 0.0186	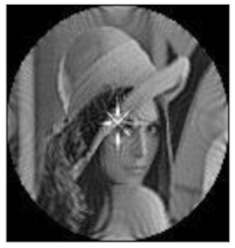 0.0223
	1	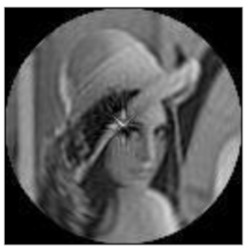 0.0162	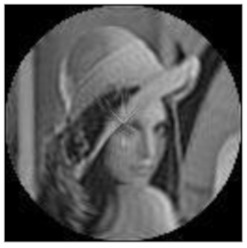 0.0152	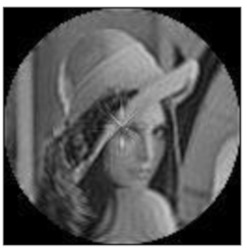 0.0147	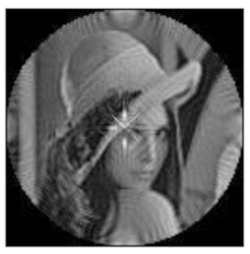 0.0156	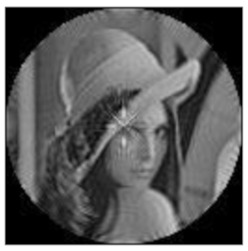 0.0189	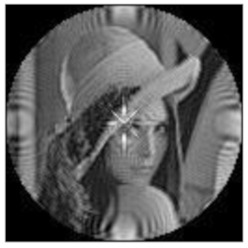 0.0301
FrRHFMs	1.2	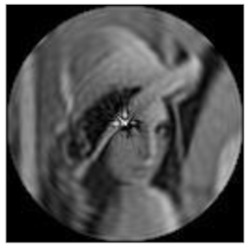 0.0430	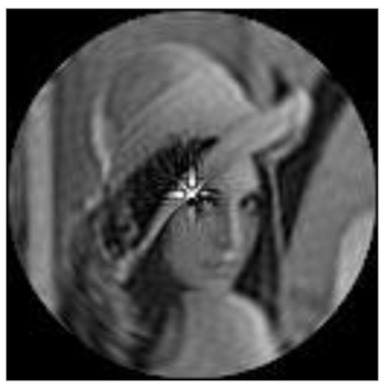 0.0470	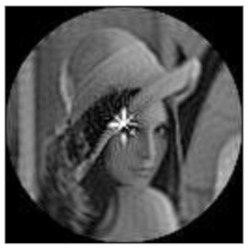 0.0550	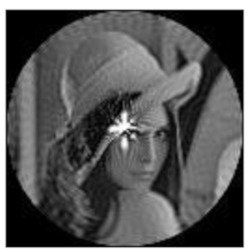 0.0627	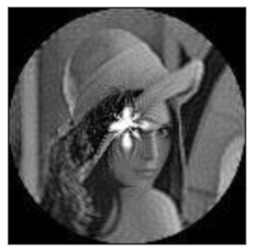 0.0734	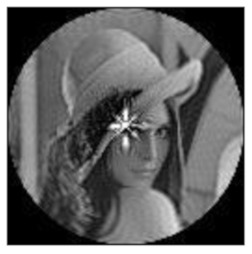 0.0871
	1.3	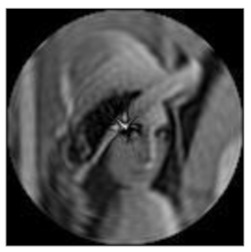 0.0507	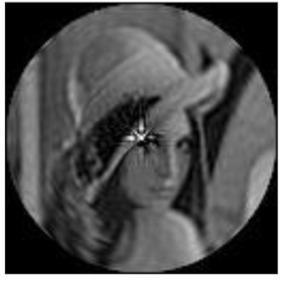 0.0521	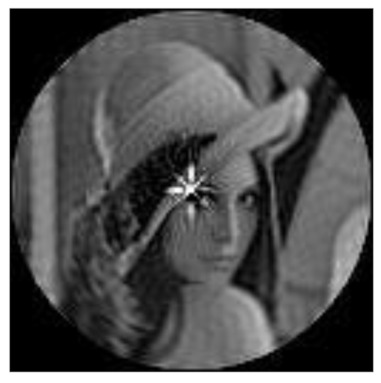 0.0559	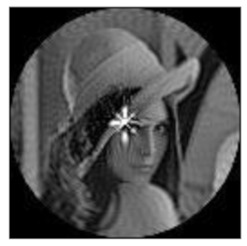 0.0609	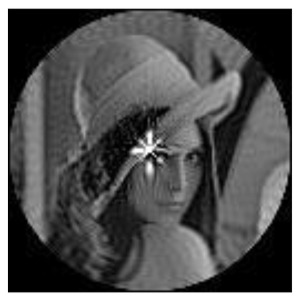 0.0672	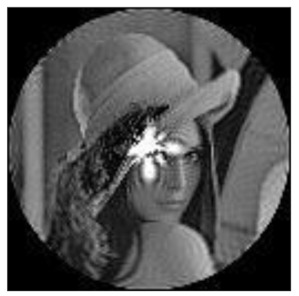 0.0753
	1.4	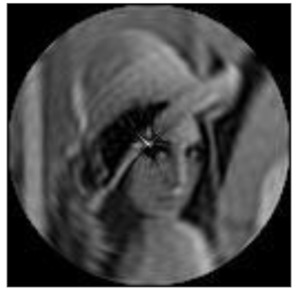 0.0632	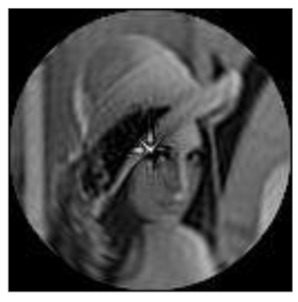 0.0628	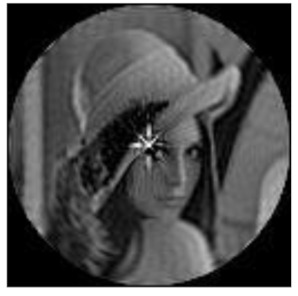 0.0641	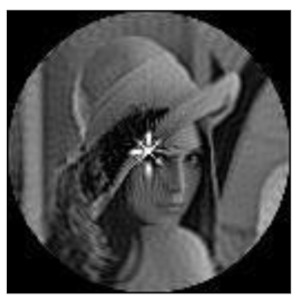 0.0665	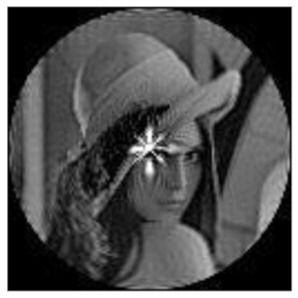 0.0707	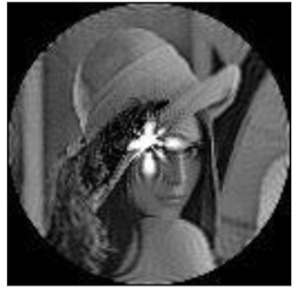 0.0761
	1	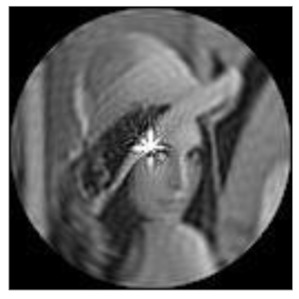 0.0659	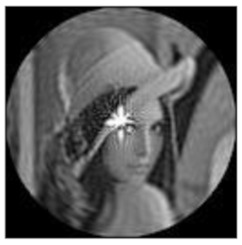 0.0857	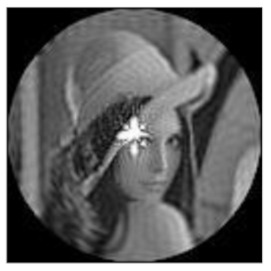 0.1104	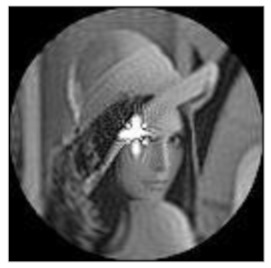 0.1419	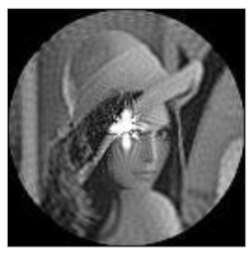 0.2023	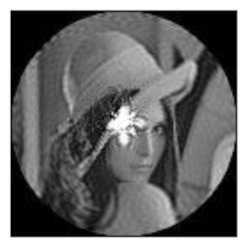 0.2805
FrEMs	1.3	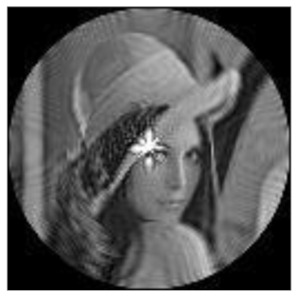 0.0468	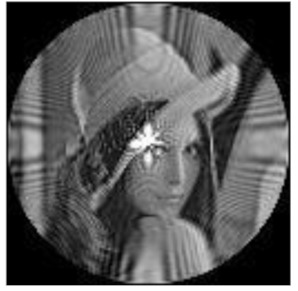 0.0776	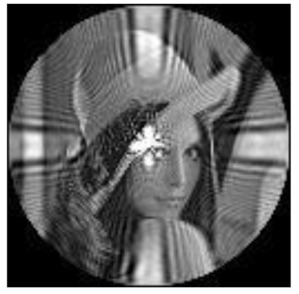 0.1380	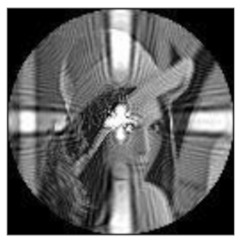 0.2237	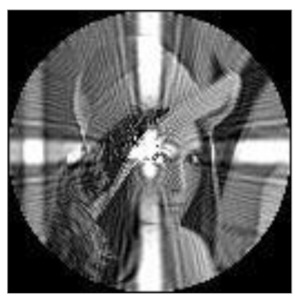 0.3363	
	1.4	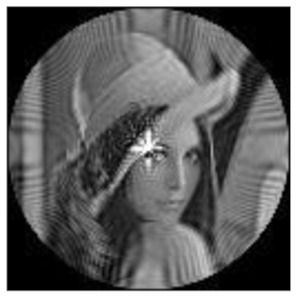 0.0406	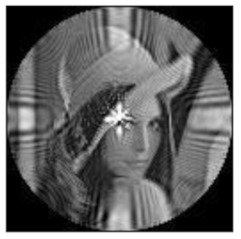 0.0782	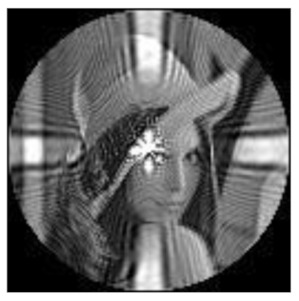 0.1359	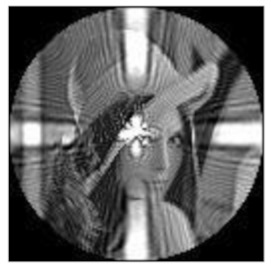 0.2169	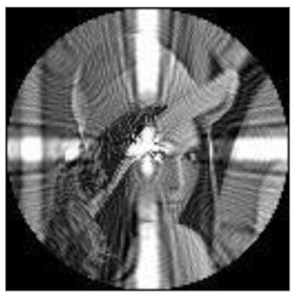 0.3229	
	1.5	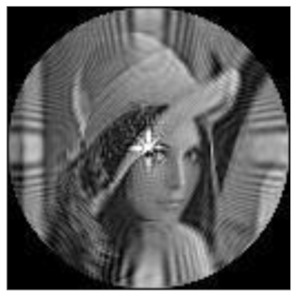 0.0438	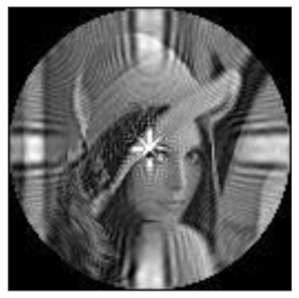 0.0813	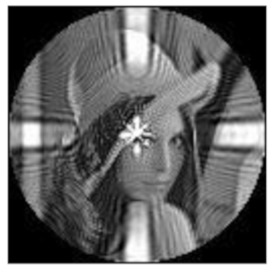 0. 1411	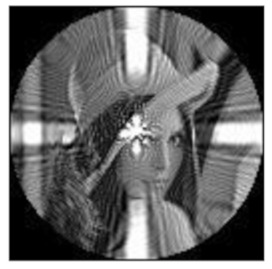 0.2227	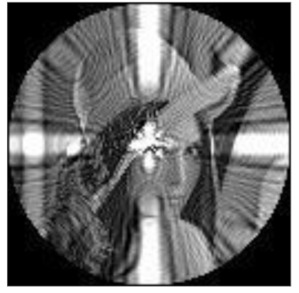 0.3271	
	1	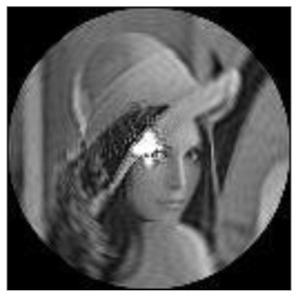 0.1571	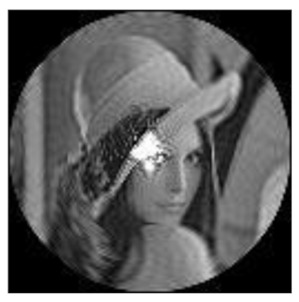 0.2580	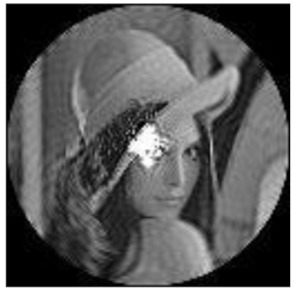 0.4266	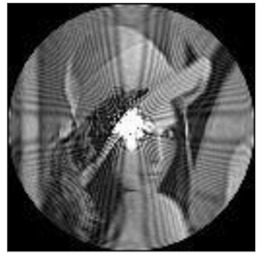 0.6839	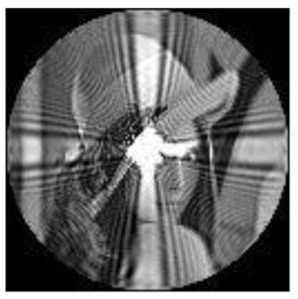 1.1081	

**Table 4 sensors-21-01544-t004:** Amplitudes of FrPHFMs of rotated Lena at t=0.7.

Attack	$\left|FP_{0,1}\right|$	$\left|FP_{0,2}\right|$	$\left|FP_{0,3}\right|$	$\left|FP_{1,1}\right|$	$\left|FP_{1,2}\right|$	$\left|FP_{1,3}\right|$	$\left|FP_{2,1}\right|$	$\left|FP_{2,2}\right|$	$\left|FP_{2,3}\right|$
No attack	13.2636	6.7878	8.1325	2.6929	7.8478	4.2117	9.8186	6.1165	2.6960
Rotation 5°	13.2801	6.8185	8.1470	2.6878	7.8399	4.2073	9.8177	6.0874	2.6973
Rotation 15°	13.2447	6.8318	8.1289	2.6875	7.8376	4.2050	9.8395	6.0204	2.7022
Rotation 25°	13.1917	6.7881	8.0857	2.6859	7.8321	4.2021	9.8409	6.0223	2.7807
Rotation 35°	13.2086	6.7705	8.0581	2.6857	7.8443	4.2037	9.8415	6.0362	2.7580
Rotation 45°	13.2480	6.7773	8.0930	2.6916	7.8296	4.2101	9.8157	6.0943	2.7395

**Table 5 sensors-21-01544-t005:** Amplitudes of FrPHFMs of rotated Lena at t=0.8.

Attack	$\left|FP_{0,1}\right|$	$\left|FP_{0,2}\right|$	$\left|FP_{0,3}\right|$	$\left|FP_{1,1}\right|$	$\left|FP_{1,2}\right|$	$\left|FP_{1,3}\right|$	$\left|FP_{2,1}\right|$	$\left|FP_{2,2}\right|$	$\left|FP_{2,3}\right|$
No attack	13.2636	6.7878	8.1325	5.0694	8.8202	4.0069	8.7714	4.6381	2.2499
Rotation 5°	13.2801	6.8185	8.1470	5.0631	8.8109	4.0025	8.7718	4.6095	2.2366
Rotation 15°	13.2447	6.8318	8.1289	5.0631	8.8080	4.0005	8.7888	4.5448	2.2606
Rotation 25°	13.1917	6.7881	8.0857	5.0638	8.7986	3.9994	8.7830	4.5582	2.3252
Rotation 35°	13.2086	6.7705	8.0581	5.0600	8.8145	3.9983	8.7897	4.5685	2.3052
Rotation 45°	13.2480	6.7773	8.0930	5.0671	8.8034	4.0062	8.7640	4.6223	2.2934

**Table 6 sensors-21-01544-t006:** Amplitudes of FrPHFMs of rotated Lena at t=0.9.

Attack	$\left|FP_{0,1}\right|$	$\left|FP_{0,2}\right|$	$\left|FP_{0,3}\right|$	$\left|FP_{1,1}\right|$	$\left|FP_{1,2}\right|$	$\left|FP_{1,3}\right|$	$\left|FP_{2,1}\right|$	$\left|FP_{2,2}\right|$	$\left|FP_{2,3}\right|$
No attack	13.2636	6.7878	8.1325	6.7837	9.4405	3.7309	7.2897	3.1317	2.0057
Rotation 5°	13.2801	6.8185	8.1470	6.772	9.4300	3.7279	7.2929	3.1016	2.0008
Rotation 15°	13.2447	6.8318	8.1289	6.7768	9.4267	3.7258	7.3040	3.0402	2.0283
Rotation 25°	13.1917	6.7881	8.0857	6.7782	9.4157	3.7261	7.2892	3.0654	2.0700
Rotation 35°	13.2086	6.7705	8.0581	6.7717	9.4330	3.7227	7.3037	3.0755	2.0502
Rotation 45°	13.2480	6.7773	8.0930	6.7796	9.4248	3.7318	7.2795	3.1212	2.0468

**Table 7 sensors-21-01544-t007:** Amplitudes of FrPHFMs of scaled Lena at t=0.7.

**Attack**	$\left|FP_{0,1}\right|$	$\left|FP_{0,2}\right|$	$\left|FP_{0,3}\right|$	$\left|FP_{1,1}\right|$	$\left|FP_{1,2}\right|$	$\left|FP_{1,3}\right|$	$\left|FP_{2,1}\right|$	$\left|FP_{2,2}\right|$	$\left|FP_{2,3}\right|$
No attack	13.2636	6.7878	8.1325	2.6929	7.8478	4.2117	9.8186	6.1165	2.6960
Scaling 0.5	13.1716	6.7492	8.1484	2.4992	7.6278	4.2069	9.9157	6.2099	2.7372
Scaling 0.7	13.1845	6.9783	8.0352	2.6448	8.1160	4.1109	9.6975	6.1228	2.5378
Scaling 1.25	13.2954	6.8006	8.1162	2.7297	7.8780	4.1928	9.8435	6.0695	2.6921
Scaling 1.5	13.2632	6.8049	8.0928	2.7462	7.8986	4.2022	9.8124	6.0241	2.7219

**Table 8 sensors-21-01544-t008:** Amplitudes of FrPHFMs of scaled Lena at t=0.8.

**Attack**	$\left|FP_{0,1}\right|$	$\left|FP_{0,2}\right|$	$\left|FP_{0,3}\right|$	$\left|FP_{1,1}\right|$	$\left|FP_{1,2}\right|$	$\left|FP_{1,3}\right|$	$\left|FP_{2,1}\right|$	$\left|FP_{2,2}\right|$	$\left|FP_{2,3}\right|$
No attack	13.2636	6.7878	8.1325	5.0694	8.8202	4.0069	8.7714	4.6381	2.2499
Scaling 0.5	13.1761	6.7492	8.1484	4.8723	8.6258	4.0048	8.8911	4.7972	2.3052
Scaling 0.7	13.1845	6.9783	8.0352	5.0677	9.0374	3.9182	8.6079	4.6351	2.1025
Scaling1.25	13.2954	6.8006	8.1162	5.1003	8.8415	3.9868	8.7910	4.5856	2.2358
Scaling 1.5	13.2632	6.8049	8.0928	5.1172	8.8581	3.9971	8.7497	4.5346	2.2736

**Table 9 sensors-21-01544-t009:** Amplitudes of FrPHFMs of scaled Lena at t=0.9.

**Attack**	$\left|FP_{0,1}\right|$	$\left|FP_{0,2}\right|$	$\left|FP_{0,3}\right|$	$\left|FP_{1,1}\right|$	$\left|FP_{1,2}\right|$	$\left|FP_{1,3}\right|$	$\left|FP_{2,1}\right|$	$\left|FP_{2,2}\right|$	$\left|FP_{2,3}\right|$
No attack	13.2636	6.7878	8.1325	6.7837	9.4405	3.7309	7.2897	3.1317	2.0057
Scaling 0.5	13.1761	6.7492	8.1484	6.6378	9.2852	3.7349	7.4022	3.3393	2.0626
Scaling 0.7	13.1845	6.9783	8.0352	6.7760	9.6124	3.6438	7.0870	3.1491	1.8804
Scaling 1.25	13.2954	6.8006	8.1162	6.8009	9.4509	3.7117	7.3133	3.0747	1.9787
Scaling 1.5	13.2632	6.8049	8.0928	6.8111	9.4619	3.7229	7.2655	3.0225	2.0219

**Table 10 sensors-21-01544-t010:** Moment sets used in object recognition and the number of moments in moment sets.

Moment	Conjugated Moment	Final Moment Set	Number of Moments
FrPHFMs	$\bar{FP_{n,m}^{\left(t\right)}}=FP_{n,-m}^{\left(t\right)}$	$S_{\mathrm{FrPHFMs}}=\left\{FP_{nm}^{\left(t\right)},n+\left|m\right|\leq K,m\geq 0,m\neq 4i,i\in Z\right\}$	${\left|S\right|}_{\mathrm{FrPHFMs}}=\left\{\begin{array}{l} \frac{3K^{2}+6K}{8}\;,\;K=4i\;\mathrm{or}\;K=4i+2\\ \frac{3K^{2}+6K-1}{8},\;K=4i+1\;\\ \frac{3K^{2}+6K+3}{8},\;K=4i+3\; \end{array}\right.$
FrRHFMs	$\bar{FM_{n,m}^{\left(t\right)}}=FM_{n,-m}^{\left(t\right)}$	$S_{\mathrm{FrRHFMs}}=\left\{FM_{nm}^{\left(t\right)},n+\left|m\right|\leq K,m\geq 0,m\neq 4i,i\in Z\right\}$	${\left|S\right|}_{\mathrm{FrRHFMs}}=\left\{\begin{array}{l} \frac{3K^{2}+6K}{8},\;K=4i\;\mathrm{or}\;K=4i+2\\ \frac{3K^{2}+6K-1}{8},\;K=4i+1\\ \frac{3K^{2}+6K+3}{8},\;K=4i+3 \end{array}\right.$
FrPCETs	$\bar{FM_{n,m}^{\left(t\right)}}=FM_{-n,-m}^{\left(t\right)}$	$S_{\mathrm{FrPCETs}}=\left\{FM_{nm}^{\left(t\right)},\left|n\right|+\left|m\right|\leq K,m\geq 0,m\neq 4i,i\in Z\right\}$	${\left|S\right|}_{\mathrm{FrPCETs}}=\left\{\begin{array}{l} \frac{3K^{2}+3K}{4},\;K=4i\;\mathrm{or}\;K=4i+3\\ \frac{3K^{2}+3K-2}{4},\;K=4i+1\;\mathrm{or}\;K=4i+2 \end{array}\right.$
FrPCTs	$\bar{FM_{n,m}^{\left(t\right)}}=FM_{n,-m}^{\left(t\right)}$	$S_{\mathrm{FrPCTs}}=\left\{FM_{nm}^{\left(t\right)},n+\left|m\right|\leq K,m\geq 0,m\neq 4i,i\in Z\right\}$	${\left|S\right|}_{\mathrm{FrPCTs}}=\left\{\begin{array}{l} \frac{3K^{2}+6K}{8},\;K=4i\;\mathrm{or}\;K=4i+2\\ \frac{3K^{2}+6K-1}{8},\;K=4i+1\\ \frac{3K^{2}+6K+3}{8},\;K=4i+3 \end{array}\right.$
FrPSTs	$\bar{FM_{n,m}^{\left(t\right)}}=FM_{n,-m}^{\left(t\right)}$	$S_{\mathrm{FrPSTs}}=\left\{FM_{nm}^{\left(t\right)},n+\left|m\right|\leq K,m\geq 0,m\neq 4i,i\in Z\right\}$	${\left|S\right|}_{\mathrm{FrPSTs}}=\left\{\begin{array}{l} \frac{3K^{2}}{8},\;K=4i\\ \frac{3K^{2}-3}{8},\;K=4i+1\;\mathrm{or}\;K=4i+3\\ \frac{3K^{2}-4}{8},\;K=4i+2 \end{array}\right.$
FrZMs	$\bar{FM_{n,m}^{\left(t\right)}}=FM_{n,-m}^{\left(t\right)}$	$S_{\mathrm{FrZMs}}=\left\{\begin{array}{l} FM_{nm}^{\left(t\right)},n-\left|m\right|=\mathrm{even},\left|m\right|\leq n\leq K,\\ m\geq 0,m\neq 4i,i\in Z \end{array}\right\}$	${\left|S\right|}_{\mathrm{FrZMs}}=\left\{\begin{array}{l} \frac{3K^{2}+8K}{16},\;K=4i\\ \frac{3K^{2}+10K+3}{16},\;K=4i+1\\ \frac{3K^{2}+8K+4}{16},\;K=4i+2\\ \frac{3K^{2}+10K+7}{16},\;K=4i+3 \end{array}\right.$

**Table 11 sensors-21-01544-t011:** CCP (%) comparison of different fractional-order continuous orthogonal moments under various attacks.

Number of Moments	Attack	FrPHFMs	FrRHFMs	FrPCETs	FrPCTs	FrPSTs	FrZMs
5	No attack	100.00	100.00	100.00	100.00	100.00	100.00
	JPEG Compression 10	99.68	97.16	96.52	93.95	93.79	96.52
	JPEG Compression 20	100.00	100.00	100.00	99.58	100.00	100.00
	JPEG Compression 50	100.00	100.00	100.00	100.00	100.00	100.00
	JPEG Compression 80	100.00	100.00	100.00	100.00	100.00	100.00
	Wiener filtering 2 × 2	100.00	100.00	100.00	100.00	99.89	99.82
	Wiener filtering 4 × 4	100.00	100.00	99.72	99.63	99.74	99.65
	Gaussian filtering 2 × 2	100.00	97.58	100.00	95.42	93.16	100.00
	Gaussian filtering 4 × 4	100.00	97.58	100.00	95.42	93.16	100.00
	Unsharp filtering 0.2	100.00	100.00	100.00	100.00	100.00	100.00
	Unsharp filtering 0.4	100.00	100.00	99.36	100.00	100.00	100.00
	Disk filtering 2	99.76	99.43	98.10	99.95	98.75	98.63
	Disk filtering 4	98.23	96.89	93.42	92.58	97.05	94.00
13	No attack	100.00	100.00	100.00	100.00	100.00	100.00
	JPEG Compression 10	99.85	99.21	98.42	98.79	98.68	99.74
	JPEG Compression 20	100.00	100.00	100.00	100.00	100.00	100.00
	JPEG Compression 50	100.00	100.00	100.00	100.00	100.00	100.00
	JPEG Compression 80	100.00	100.00	100.00	100.00	100.00	100.00
	Wiener filtering 2 × 2	100.00	100.00	100.00	100.00	100.00	100.00
	Wiener filtering 4 × 4	100.00	100.00	100.00	100.00	100.00	100.00
	Gaussian filtering 2 × 2	100.00	99.68	97.95	99.05	97.95	100.00
	Gaussian filtering 4 × 4	100.00	99.68	97.95	99.05	97.95	100.00
	Unsharp filtering 0.2	100.00	100.00	100.00	100.00	100.00	100.00
	Unsharp filtering 0.4	100.00	100.00	100.00	100.00	100.00	100.00
	Disk filtering 2	100.00	100.00	100.00	100.00	100.00	100.00
	Disk filtering 4	98.31	97.95	95.26	94.89	96.32	96.78

## Data Availability

Data available on reasonable request from the corresponding author.

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
