# Peer review of "Invariant Image Representation Using Novel Fractional-Order Polar Harmonic Fourier Moments"

_sensors, 2021, doi:10.3390/s21041544_

Round 1
Reviewer 1 Report
I have analyzed the manuscript about the extension of the functions used to image representation to fractional ones and their applications in real situations. The results were very positive about the quality concerning the use of these functions. The manuscript is well written, and the subject is interesting and useful for the researches on this subject. For this reason, I recommend for publication the present article.
Reviewer 2 Report
The authors introduce a fractional-order extension to their Polar Harmonic Fourier Moments method. It is a natural and interesting step forward in this line of research which has many important applications.
My main concern is that the results do not seem to support/show all the advantages claimed in the abstract and the conclusions. In the first study (with various values of t) (Table 2). The non-fractional case (t=1), seems to give the best results (with order n=60). Non-fractional cases only seem to be better at very high orders, something which does not seem to provide any practical advantage. (Unless the authors show some case where n>70 is required).
On the other hand, the comparison with other non-fractional moments, probably requires a more in-depth evaluation, as it may be the case, that not all moments have the same optimal "t" value. It would be good to have as a reference in these low-order cases (which are the most useful ones), the reference of the t=1 case (at least for the Polar Harmonic Fourier Moment).
Furthermore, there are some important pieces of information that I missed. For instance, in addition to the maximum-order considered, it would be good to have the value of K in each case, as the number of moments considered would be (K+1)^2. This is important for many applications. For instance, how many moments are used to obtain the representation of an image with 128x128 pixels? In some of the final results, the value of t used in all cases should be indicated.
Some minor/detailed comments:
1 ) Fig. 1 - Group these 4 figures in a 2x2 figure. Large figures are not needed for this.
2 ) Fig. 3 - Arrange better the images (3 rows should be enough).
3 ) Table 2 - Maybe a grayscale image of the value of eps for each nmax,t would be helpful.
4 ) Table 2- Indicate what happens with t< 0.7? t>1.2?
5 ) Comment if some other metric (other than MSRE) (for instance, mean absolute error) could be used, and in that case, if the result would vary. Another variation, may be to restrict the evaluation area to a particular region (avoiding for instance the central point or the points away from some distance). Some of these metrics may be better for the evaluated cases.
Reviewer 3 Report
The authors incorporate fractional-order exponents in polar harmonic Fourier moments. The expression (7) is an interesting achievement.
Equation (10) is not derived, this is an important result!
Why is not (12) presented in Section 2.1? This would place moments computation and their use for synthesis together.
Equation (13) has errors. (15) is not derived, in addition, the scaling shown alters the norm and the meaning of "the normalized image function" is unclear.
The last paragraph of Section 3 is not clear and makes very strong claims with few data, references, or results supporting them.
The data in Table 2 does not seem to agree with their description in the paragraph after (17).
The results could be synthesized in a smaller quantity of graphs to improve readability and comparisons. The authors should take their time to improve the presentations of results.
Some minor comments:
The expressions (3) and (7) could be presented in a better form, placing the sin/cos arguments within parentheses.
In the sentence before (11) and others n(n>=0) could be simply n>=0 the same applies to |m|.
The presentation of figures 1 and 2 is bad and difficult to follow, also the "change rate" is not defined.
(16) repeats (12)
Reviewer 4 Report
The paper is very interesting. Authors propose new approache to PHFMs, which can only take integer numbers. They method bases on an extending of PHFMs to FrPHFMs by means of modification of their radial polynomials. In this case the FrPHFMs not only maintain the orthogonality, rotation invariance and scale invariance of integer-order PHFMs. New radial polynomials have stronger image description ability and are superior to integer-order PHFMs and other fractional-order continuous orthogonal moments. These apply to image reconstruction, noise resistance and object recognition.
Carried out analyses apply several grayscale images. In my opinion it is a limitation. Obtained results are correct. I have one doubt. I think that the authors should carry out analyses for color images.
The paper has a well overall structure, divided in five chapter, with an informative conclusion with a summary of the research results. But the Conclusion has to be enhanced.
Also the paper doesn't include: Author Contributions and a declaration about Conflicts of Interest.
Round 2
Reviewer 3 Report
Nothing to add in this roundd.
